# Effects of short birth interval on different forms of child mortality in Bangladesh: Application of propensity score matching technique with inverse probability of treatment weighting

**Mohammad Zahidul Islam[1,2], Md. Mostafizur Rahman[2], Md. Nuruzzaman Khan** [1]*

1 Department of Population Science, Jatiya Kabi Kazi Nazrul Islam University, Trishal, Mymensingh, Bangladesh, 2 Department of Population Science and Human Resource Development, Rajshahi University, Rajshahi, Bangladesh

* mdnuruzzaman.khan@uon.edu.au

**Data Availability Statement:** The data that support the findings of this study are available from The DHS Program and legal restrictions apply to the

## Abstract

### Background

The prevalence of Short Birth Interval (SBI) is higher in Low- and Middle-Income countries (LMICs), including Bangladesh. Previous studies in LMICs have estimated the effects of SBI on child mortality by comparing two unequal groups of mothers based on their socio-economic status. This approach may lead to overestimation or underestimation of the true effect of birth interval on child mortality, particularly when sample sizes are relatively small.

### Objective

We determined the effects of SBI on several forms of child mortality in Bangladesh by comparing two equal groups created by applying the propensity score matching technique.

### Methods

This study analyzed data from 5,941 mothers and 1,594 health facilities extracted from the 2017/18 Bangladesh Demographic and Health Survey and the 2017 Bangladesh Health Facility Survey. The exposure variable was SBI (defined as the interval between two subsequent births <33 months: yes, no), while the outcome variables were neonatal mortality (defined as mortality within 28 days of birth: yes, no), infant mortality (defined as mortality within 1 year of birth: yes, no), and under-five mortality (defined as mortality within 5 years of birth: yes, no). Multilevel Poisson regression based on inverse probability treatment weights was used to determine the association between exposure and outcome variables.

### Results

The prevalence rates of neonatal, infant, and under-five mortality were 48.8, 30.8, and 23.1 per 1000 live births, respectively. Newborns of SBI mothers were found to have a 63%

availability of these data, which were used under license for the current study, and so are not publicly available. Data are, however, available from The DHS Program (https://dhsprogram.com) upon submission of research proposal and/or reasonable request. To access the relevant data used in this study, interested authors need to submit research proposal based on the 2017/18 BDHS and 2017 BHFS. The variables used in this study are directly available in the datasets, so interested authors can easily find out the variables for analysis. No special privileges is required for this and the authors of this study did not have this.

**Funding:** The author(s) received no specific funding for this work.

**Competing interests:** The authors have declared that no competing interests exist.

**Abbreviations:** aPR, adjusted prevalence ratio; BDHS, Bangladesh Demographic and Health Survey; BHFS, Bangladesh Health Facility Survey; DAG, Directed Acyclic Graph; DHF, district health facilities; LMICs, low- and middle-income countries; MCWC, mother and child welfare centre; MDGs, Millennium Development Goals; SBI, short birth interval; SDGs, Sustainable Development Goals.

higher likelihood of neonatal mortality (aPR, 1.63; 95% CI, 1.08–2.46) compared to new-borns of non-SBI mothers. Furthermore, the prevalence of infant mortality and under-five mortality was 1.45 times higher (aPR, 1.45; 95% CI, 1.01–2.08) and 2.82 times higher (aPR, 2.82; 95% CI, 2.16–3.70), respectively, among babies born in a short interval of their imme-diately preceding sibling as compared to babies born in a normal interval of their immedi-ately preceding sibling.

## Conclusions

Findings of this study indicate that SBI is an important predictor of child mortality. Conse-quently, around 1 million children born in a short interval every year in Bangladesh are at risk of dying before reaching their fifth birthday. This indicates a challenge for Bangladesh to achieve the SDG 3 target to reduce neonatal and under-five mortality to 12 and 25 deaths per 1000 live births, respectively. Hence, awareness-building programs about the adverse effects of SBI and strengthening existing healthcare facilities are important.

## Introduction

The world has observed a remarkable reduction in under-five mortality during the Millen-nium Development Goals (MDGs) period from 2000 to 2015 [1, 2]. The global under-five mortality rate declined to 43 per 1000 live births in 2015 from 75 per 1000 live births in 2000 [2]. This was equivalent to around a 32% drop in 15 years, with 1 in 13 children dying before age 5 in 2000 compared to 1 in 23 in 2015 [3]. However, this progress was insufficient to achieve the MDG target of a two-thirds reduction of under-five mortality. By the end of the MDGs in 2015, around 5.9 million under-five deaths occurred worldwide, with an estimated 16,000 deaths every day [3]. The progress was insufficient in sub-Saharan Africa and South Asia, where around 81% of the global under-five deaths occurred in 2015 [1, 3]. Around half of these deaths occur within 28 days of life, with a 5% increase from 40% in 1990. In this cir-cumstance, in 2015, world leaders set a Sustainable Development Goals (SDGs) target to reduce neonatal and under-five mortality to 12 and 25 per 1000 live births, respectively, by 2030 [4]. However, in 2022, around the middle time of the SDGs' period, 17 and 37 neonatal and under-five deaths were recorded per 1000 live births, respectively [5]. Importantly, the rate of reduction for the first half of the SDGs period, from 2015–2021, was not adequate to achieve the SDGs' child mortality targets by 2030 and was even lower than the rates of reduc-tion for the equal number of years in the MDGs' period in respect to achieving the relevant MDGs' targets [5]. Therefore, this is considered an ongoing public health threat, particularly in sub-Saharan Africa and South Asia, although interventions at the maternal and health facil-ity level can help prevent almost all of these deaths [3].

Bangladesh, a low- and middle-income country (LMICs), is the second major contributor to the South Asian under-five deaths toll [6], with current estimated neonatal and under-five mortality rates at 17 (95% CI: 15–20) and 29 (95% CI: 26, 32) per 1000 live births, respectively [7]. Bangladesh achieved the MDGs target for under-five mortality, making it one of the few countries to do so. With around 4 million yearly births, it has the second-highest number of births in South Asia after Pakistan [8]. The population momentum has created a bulge in the younger generation in Bangladesh, as well as other South Asian countries and LMICs [9]. Therefore, it is expected that the present number of births will continue to grow. This indicates

Bangladesh will continue contributing to higher numbers of under-five deaths and unless preventive action is taken, the country will not meet the SDGs targets by 2030 [10]. Bangladesh needs to strengthen its healthcare facilities to ensure effective maternal healthcare services use and address pregnancy-related complications and deaths [10]. Population-level challenges, including early marriage, immunization coverage, maternal and under-five children undernutrition, violence against children and improvement of water and sanitation, must also be prioritised in existing policies and programs [11]. A similar recommendation can be made for other LMICs [11].

Short Birth Interval (SBI), defined as an interval of less than 33 months between two subsequent births, has been found to be linked to multiple factors affecting under-five mortality, including early marriage, pregnancy complications, and maternal and under-five children's nutritional disorders. This linkage may arise in direct and indirect ways: (i) women having these characteristics, such as early marriage, prefer births in short intervals, and (ii) short interval births increase adverse outcomes, such as pregnancy complications and maternal and under-five nutrition. The associations are mediated by other factors, including the disadvantaged socioeconomic conditions of the mothers, previous child survival status, and the distance from home to nearby clinics [12–14]. These linkages indicate a potential association between SBI and under-five mortality. A systematic review and meta-analysis of 51 studies conducted in LMICs by the authors of this current study also found a significant linkage of SBI with stillbirths, neonatal mortality, infant mortality, and under-five mortality [15]. However, the review included articles based on secondary cross-sectional data with a relatively small sample size for mortality events and had at least two notable limitations. First, the estimated mortality was produced by comparing two unequal groups with respect to socio-demographic factors, which are independent predictors of child mortality. The influence of health facility-level factors, also an important predictor of child mortality, was mostly ignored in the published articles, which is the second major limitation [13]. Furthermore, some other notable limitations of the published papers were the consideration of different intervals to define SBI, a small list of confounding factors, and the analysis of a sample of rural or urban areas [16–19]. A recent study conducted in Ethiopia attempted to address the limitation of comparing socio-demographically unequal groups; however, the authors of this paper did not address the health facility-level factors [20]. The present study aims to determine the effects of SBI on neonatal, infant, and under-five mortality. The effects were estimated by analysing nationally representative data so that mothers with similar socio-demographic factors could be compared while adjusting for potential confounders.

## Methods

### Study design and sampling

The 2017/18 Bangladesh Demographic and Health Survey (BDHS) and the 2017 Bangladesh Health Facility Survey (BHFS) data were linked and analysed in this study. Both are nationally representative surveys conducted as part of the Demographic and Health Survey Program of the USA. The details of these surveys have been published elsewhere [21, 22], with key aspects of these surveys are presented here.

The 2017/18 BDHS was the seventh round of BDHS conducted by a government-owned Bangladeshi research firm, the National Institute of Population Research and Training, along with a private research firm, Mitra and Associates. The Demographic and Health Survey Program of the USA provided the overall supervision. The international development partners of Bangladesh, including the UNFPA and USAID, provided the financial and technical support to conduct this survey.

The reproductive-aged married women were the main sample in this survey, included from the nationally representative selected households. The conditions applied for women to be included in the survey were (i) usual residents of the selected household or (ii) stayed the most recent night before the survey at the selected households. The households were selected in two stages. In the first stage, a total of 675 clusters were selected randomly from a list of 293,579 clusters of Bangladesh generated by the Bangladesh Bureau of Statistics as part of the 2011 Bangladesh National Population Census. Three clusters were excluded due to flood and the remaining 672 clusters were included in the survey. A fixed number of 30 households from each cluster was selected at the second stage of sampling. This produced a list of 20,160 households, of which the survey was conducted in 19,457 households with an over 96% inclusion rate. There were 20,376 eligible women in these selected households, and 20,127 women were interviewed with a response rate of 98.8%.

The 2017 BHFS used a list of 19,811 registered health facilities as a sampling frame generated by the Ministry of Health and Family Welfare. A total of 1,600 health facilities were selected from this list and 1,524 health facilities were finally included in the survey. The selection was made following the census of the district health facilities (DHF) and mother and child welfare centre (MCWC) and stratified random sampling of other healthcare facilities of the government, private, and non-governmental organizations [22].

## Study sample

The BDHS recorded birth interval data for 5,941 women that we analysed in this study. The inclusion criteria for this sample were: (i) women who had at least two pregnancies, of which the most recent one ended with live birth within five years of the survey date, (ii) the second most recent pregnancy ended with live birth or termination, and (ii) the end dates of pregnancies and the interval before the most recent live birth were recorded.

## Outcome variable

We considered three outcome variables: neonatal mortality (death occurred within 1 month of live birth), infant mortality (deaths occurred within one year of live birth) and under-five mortality (death occurred within five years of live births). The BDHS recorded these mortality data by asking women whether they had any live birth within five years of the survey date and the survival status of their respective children. In the occurrence of more than one live birth within five years, the survival status data were collected for every child. These data were then categorized using the WHO guidelines to generate the outcome variables. The exclusion criteria was multiple pregnancies because of their different death patterns, and only singleton pregnancies were analysed [23, 24].

## Explanatory variables

The primary explanatory variable was SBI, defined as an interval of <33 months between the two most recent births as recommended by the WHO [25]. The BDHS recorded this data in months by subtracting the date of birth of the most recent child from the date of birth or termination of the second most recent child. Relevant data were recorded during the survey by seeing birth registration reports or immunization card. If none were available, the mothers were requested to recall their memories by referring to memorable events like floods or the national election.

We conducted a comprehensive literature search in Medline, Embase, Web of Science, CINHAL, and Google Scholar to identify the key factors to study the association between birth interval and under-five mortality. For this, special attention was paid to the studies conducted

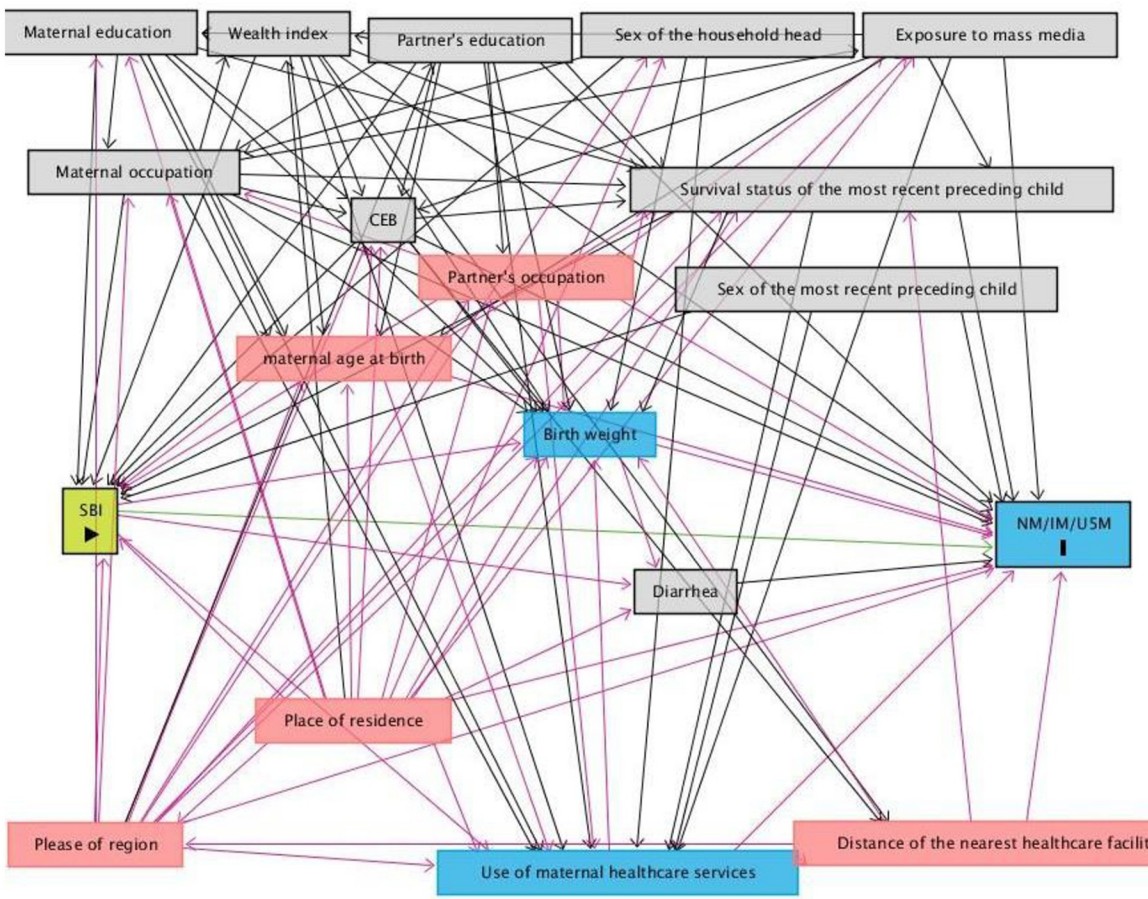

**Fig 1. Direct acyclic graph used to select controlling variables to study the relationship of the short birth interval with neonatal, infant and under-five mortality. Note**: CEB, children ever born, Nm, neonatal mortality, IM, infant mortality, and U5M, under-five mortality.

in Bangladesh [16–19, 26, 27] and the studies conducted in some other LMICs [28–33]. The availability of the selected variables in the survey was then checked. Finally, the selected variables whose data were available in the survey were then included in the DAG (Directed Acyclic Graph, Fig 1), where interrelationships among the included variables were considered carefully by the expert researchers in this group (Khan MN and Rahman MM). This produced a list of confounding factors to be studied in the relationship, which we considered in our analysis. The factors were maternal age at birth, maternal education, maternal formal employment status, partner's educational status, sex of the household's head, total children ever born, and exposure to mass media. The sex of the deceased child's sibling was also considered.

## Statistical analysis

Descriptive statistics, frequency and mean, was used to describe the characteristics of the respondents. Considering the relatively small data for neonatal, infant and under 5 mortality (rare events), we applied the propensity score matching technique with inverse probability of treatment weighting to estimate the unbiased effect of SBI on each outcome [20]. This method is the best because compared to other propensity score methods, it preserves the sample size of the original data and provides an appropriate estimation of the variance of the main effect

and maintains an appropriate type I error rate [34–36]. Given observed baseline covariates, the propensity score assigned the probability of treatment among the observations after adjusting the confounders due to non-random treatment [20]. Inverse probability of treatment weighting aided in reducing the observations using matching for the confounder adjustment and providing weights to the entire study sample depending on the conditional probability of treatment [34, 37–39]. Details of how the propensity score with inverse probability of treatment weighting works was given elsewhere [20].

Since the BDHS employed a complex two-stage, stratified, clustered random sampling procedure, we applied the sampling weights to adjust the non-proportional allocation of sample participants to different clusters [21]. We used the multilevel Poisson regression model to estimate the treatment effect (SBI) on outcomes. Final estimates of the outcome variables were incorporated with the product of sampling weight and the stabilized weight by the inverse probability of treatment weighting. The incorporation of both weights is required to estimate the unbiased treatment effects which can be generalized to the original survey population. Effects were expressed in adjusted prevalence ratio (aPR) with their 95% confidence interval. All the analyses were performed in the Stata version 15.10 (StataCorp Stata Statistical Software: Release V.15. College Station, Texas: StataCorp LP 2017).

## Results

### Background characteristics of the respondents

The key information about the study participants, exposure and outcome variables are presented in Table 1. The mean age of the mothers was 26 years, and the mean weight was 52 kilograms. The mean age of the child analysed was nearly 31 months. Around 26% of the total mothers analysed had short interval births. The under-five, infant and neonatal mortality rates were 48.8, 30.8 and 23.1 per 1000 live births, respectively.

### Distribution of neonatal mortality, infant mortality, and under-five mortality by background characteristics

Different distributions of the under-five, infant, and neonatal mortality are presented in Table 2. We found a higher prevalence of under-five mortality among comparatively less aged

**Table 1. Key information about the study participants, exposure, and outcome variables.**

| Demographics of mothers | Frequency (n = 5946) | Weighted percentage (95% CI) |
|---|---|---|
| **Mean age in years (mean ± SD)** | 5946 | 25.93 (±5.13) |
| Mean weight in kilograms (mean ± SD) | 5860 | 52.04 (±10.38) |
| Mean years of education (mean ± SD) | 5946 | 6.12 (±3.70) |
| **Demographics of under-five children** | | |
| Mean age in months (mean ± SD) | 5946 | 30.55 (±17.77) |
| Girls | 5946 | 47.75 (46.40–49.11) |
| **Birth interval** | | |
| Birth in short interval | 1566 | 26.36 (24.84–27.94) |
| Birth not in short interval | 4375 | 73.64 (72.06–75.16) |
| **Outcomes** | | |
| Neonatal mortality per 1000 live births | 137 | 23.1 (19.1–28.0) |
| Infant mortality per 1000 live births | 182 | 30.8 (26.1–36.2) |
| Under-five mortality per 1000 live births | 289 | 48.8 (42.7–55.7) |

SD Standard deviation, CI confidence interval.

**Table 2. Neonatal, infant and under-five mortality rates by background characteristics, BDHS 2017/18.**

| Characteristics | Neonatal mortality rate per 1000 (95% CI) | Infant mortality rate per 1000 (95% CI) | Under-5 mortality rate per 1000 (95% CI) |
|---|---|---|---|
| **Maternal age at birth** | | | |
| ≤19 | 35.4 (19.3–64.1) | 46.0 (27.5–76.0) | 135.8 (106.0–172.4) |
| 20–34 | 21.9 (17.7–27.1) | 29.6 (24.6–35.6) | 42.4 (36.3–49.5) |
| ≥35 | 24.1 (11.7–49.2) | 27.3 (13.8–53.1) | 28.3 (14.7–53.9) |
| **Mother's educational status** | | | |
| No education | 28.0 (16.8–46.1) | 35.3 (22.1–56.0) | 58.7 (39.6–86.1) |
| Primary | 26.9 (19.2–37.5) | 37.4 (28.4–49.2) | 52.4 (42.0–65.2) |
| Secondary | 19.6 (14.5–26.4) | 25.9 (20.0–33.6) | 46.3 (38.2–56.1) |
| Higher | 22.6 (11.7–43.2) | 27.2 (15.2–48.4) | 39.4 (25.1–61.2) |
| **Mother's participation in formal work** | | | |
| Yes | 29.2 (22.8–37.2) | 35.0 (28.1–43.5) | 50.1 (41.6–60.1) |
| No | 17.8 (12.9–24.5) | 27.0 (21.1–34.6) | 47.7 (39.7–57.1) |
| **Partner's educational status** | | | |
| No education | 27.9 (17.8–43.5) | 38.7 (26.9–55.4) | 56.9 (42.8–75.3) |
| Primary | 23.0 (16.7–31.7) | 30.3 (22.8–40.3) | 45.9 (36.4–57.7) |
| Secondary | 19.9 (13.4–29.4) | 29.3 (21.0–40.7) | 51.8 (40.6–65.8) |
| Higher | 23.6 (14.5–38.2) | 25.2 (15.9–39.8) | 40.2 (28.3–56.7) |
| **Sex of the household's head** | | | |
| Male | 22.3 (18.0–27.6) | 30.8 (25.8–36.7) | 50.1 (43.6–57.4) |
| Female | 27.9 (17.5–44.2) | 30.7 (19.3–48.3) | 40.9 (27.6–60.0) |
| **Total children ever born** | | | |
| ≤2 | 12.6 (0.9–1.8) | 16.7 (12.2–22.7) | 45.1 (37.0–54.9) |
| >2 | 34.4 (27.3–43.2) | 45.9 (37.7–55.7) | 52.7 (43.7–63.5) |
| **Exposure to mass media** | | | |
| Little exposed | 20.9 (15.5–28.1) | 29.5 (23.1–37.5) | 49.2 (40.2–60.2) |
| Moderate exposed | 26.8 (20.5–35.0) | 33.3 (26.1–42.2) | 49.0 (40.1–59.7) |
| Highly exposed | 15.9 (0.9–29.6) | 25.4 (16.0–40.2) | 46.7 (33.0–65.8) |
| **Sex of the deceased children sibling** | | | |
| Male | 20.9 (15.7–27.7) | 28.4 (22.3–36.3) | 50.9 (42.0–61.6) |
| Female | 25.2 (19.7–32.2) | 32.9 (26.8–40.4) | 46.8 (39.6–55.2) |

and illiterate mothers. The under-five mortality rate was also higher among mothers whose husbands had no education at the time of the survey, mothers who resided in male-headed households and mothers who had >2 children ever born. A similar distribution was reported for infant and neonatal mortality.

## Balance diagnosis

**Propensity score balance.** The balance of propensity score (PS) before and after weighting across treatment (navy blue line) and control group (red line) are presented in Fig 2. The figure shows an adequate balance of the PS distribution between the treatment group after weighting.

**Covariate balance.** The balance of the PS for the covariates included in this study before and after weighting across treatment and control groups are presented in Table 3. After weighting adjustment, standardized differences of all covariates were <1%. This indicates good balance and comparability between the women with and without SBI.

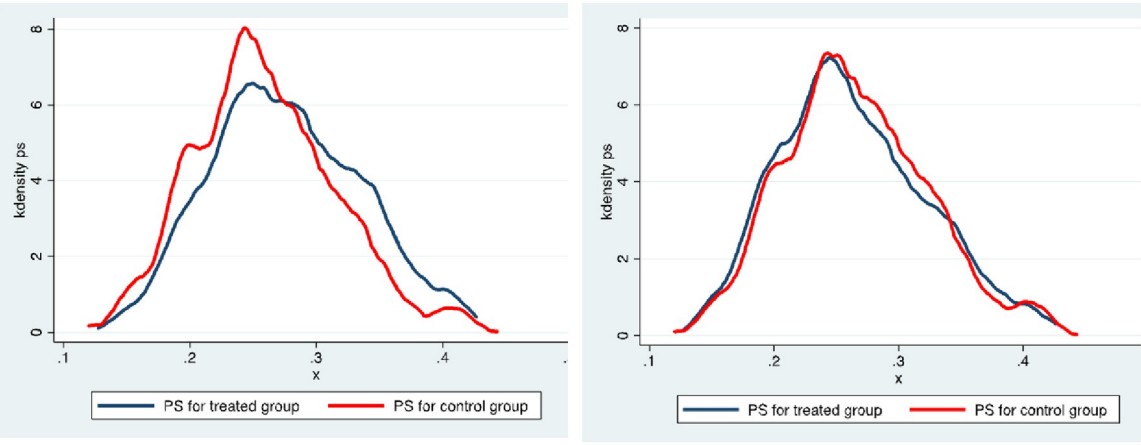

Propensity score BEFORE weighting

Propensity score AFTER weighting

**Fig 2. Balance of propensity scores (PS) before and after weighting across treatment and comparison groups.**

### Effects of short birth interval on neonatal, infant and under-five mortality

After ensuring the balance of covariates, the association of SBI with neonatal, infant and under-five mortality were estimated separately using a multi-level Poisson regression model in the weighted sample adjusted for covariates. Results are presented in Table 4. The adjusted estimated odd ratio of neonatal mortality was 63% higher (aPR,1.63; 95% CI, 1.08–2.46) among babies of the mothers who experienced SBI than the babies of the mothers who did not experience SBI. The likelihood of infant mortality was also found 45% (aPR,1.45; 95% CI, 1.01–2.08) higher among babies who were born in short intervals than their counterparts born in normal intervals. The reported effect was even strongest for under-five mortality. We found 2.82 times (aPR,2.82; 95% CI, 2.16–3.70) higher likelihood of under-five mortality among children of the mothers who had their births in short intervals as compared to their counterparts who had normal intervals of successive births.

## Discussion

We aimed to estimate the effects of SBI on neonatal, infant and under-five mortality by analyzing nationally representative data in such a way that mothers of similar socio-demographic factors were compared along with the adjustment of potential covariates. We applied the

**Table 3. Covariates PS balance before and after weighting.**

| Covariates | Mean in treated group | Mean in comparison group | Standardized differences |
|---|---|---|---|
| Maternal age at birth | 0.78 | 0.77 | 0.012 |
| Mother's educational status | 1.57 | 1.59 | -0.027 |
| Mother's participation in formal work | 0.46 | 0.47 | -0.013 |
| Partner's educational status | 1.39 | 1.43 | -0.033 |
| Sex of the household's head | 1.12 | 1.12 | -0.015 |
| Total children ever born | 1.51 | 1.48 | 0.019 |
| Exposure to mass media | 0.73 | 0.73 | -0.009 |
| Sex of the deceased child's sibling | 1.11 | 1.10 | 0.009 |

**Table 4. The effect of short birth interval on neonatal, infant and under-five mortality in Bangladesh, BDHS 2017/18.**

| Treatment variable | Neonatal mortality | | aPR[+] (95% CI) |
|---|---|---|---|
| **Short birth interval** | No (%) | Yes (%) | |
| No | 4283 (73.80) | 92 (66.71) | Ref |
| Yes | 1520 (26.20) | 46 (33.29) | 1.63 (1.08–2.46)** |
| **Short birth interval** | Infant mortality | | |
| | No (%) | Yes (%) | |
| No | 4250 (73.80) | 125 (68.48) | Ref |
| Yes | 1509 (26.20) | 58 (31.52) | 1.45 (1.01–2.08)** |
| **Short birth interval** | Under-five mortality | | |
| | No (%) | Yes (%) | |
| No | 4222 (74.87) | 150 (51.87) | Ref |
| Yes | 1417 (25.13) | 139 (48.13) | 2.82 (2.16–3.70)*** |

**Note:** [+]Models adjusted with maternal age at birth, maternal education, mother's participation in formal work, partner's educational status, sex of the household's head, total children ever born, exposure to mass media and sex of the deceased child's sibling.

**p<0.05,

***p<0.01

propensity score matching technique with inverse probability of treatment weighting after specifying the set of explanatory variables using the DAGs. The associations were determined using the multi-level Poisson regression model. We found a higher likelihood of neonatal, infant, and under-five mortality among the mothers who had a SBI compared to their counterparts with normal birth intervals. The reported associations were robust in several areas, including the use of advanced statistical modelling with an appropriate weighting and confounders selected by the DAG, and consideration of household survey data together with the health facility survey data. As far as we know, this study is the first of its kind in Bangladesh as well as in LMICs. We suggest that the SBI is a strong predictor of neonatal, infant and under-five mortality in Bangladesh.

The association reported in this study was consistent with the previous studies in LMICs, including Nigeria [28, 40, 41], Ethiopia [20, 30, 32, 42, 43], Ghana [44], India [26], and Pakistan [33]. However, our estimated prevalence ratio were lower than previous studies. One reason could be our use of WHO interval to classify SBI (<24 months) in this study rather than the intervals used by previous authors, ranging from <18 to 24 months. The methodological advantage is also an important contribution to such variation. Considering the significance of the findings, the reported associations in this study should be given priority in relevant policy reforms in LMICs and Bangladesh.

The association between the SBI and neonatal mortality can arise in several pathways. There is evidence that when mothers become pregnant in shorter intervals, they face several adverse outcomes during pregnancy and delivery. For instance, it was found that the occurrence of births in short intervals increases folate depletion/anomalies, maternal nutritional deficiencies [20, 45–47], uterine rupture, postpartum hemorrhage [48, 49], congenital anomalies [47, 49] and gestational diabetes [45, 50–52]. It is also found that the SBI increases the occurrence of cesarean delivery [52–55]. These adverse maternal health outcomes could increase the occurrence of adverse birth outcomes, such as small for gestational age [49], massive genital bleeding [56], birth asphyxia and hypoxia [52, 54], preterm birth, low birth weight,

and neonatal sepsis [45, 50–52]. All adverse outcomes could lead to neonatal mortality, mostly soon after delivery- the period when over 70% of the total neonatal deaths and most under-five deaths occurred in LMICs and Bangladesh [3, 11].

Even if the child passes this critical stage, along with the demand for extra care due to these adverse consequences during the neonatal period, they face additional challenges following their presence into infancy. For instance, the birth of two children in a closer period increases competition among siblings for care and food. This is an important direction considering the prevalence of short birth intervals is high among mothers of lower socio-demographic conditions because of their, (i) lack of awareness about the importance of longer birth intervals, (ii) inadequate knowledge of family planning services and (iii) lower use of contraception, or more likely some combinations of three. Importantly, mothers of lower socio-demographic conditions also have limited capacity to buy nutritious foods for their multiple children, which could lead to a higher prevalence of nutritional deficiency among the children. Furthermore, these limitations create demand from the children for additional food, which could reduce the mothers' capacity for infant feeding, either through the supply of formula milk or breastfeeding, the latter an important determinant of child survival [57]. However, even if the mothers manage to fulfill all of these, the duration and intensity of breastfeeding is always lower for children born in short intervals than children born in normal intervals. All these could affect infants' nutritional status and resistance to infection and may lead them to death [57–63].

These adverse consequences continue along with the children's increasing age, unless protective measures, such as breastfeeding, nutritional foods, and immunization, have been taken. However, since short interval births mainly occur among socio-economically disadvantaged mothers, it is unlikely that the mothers would take preventive measures except if they were counselled properly to do so. This needs educating mothers about the importance of childcare, particularly for the children born in shorter intervals. Healthcare facilities also need to play a major role in ensuring child healthcare services when needed and other preventive measures in a timely manner, such as immunization. However, there is evidence that children born in short intervals are less likely to receive immunization with continuity [30, 31, 43, 64, 65], mostly because of their mothers' lack of time and motivation to bring their children to the health centre [20]. This increases the occurrence of infectious diseases, including diarrhea and pneumonia. These, along with the existing adverse consequences, can increase under-five mortality.

Our findings imply that there needs to be substantial investment in health promotion to raise awareness about the impotence of longer interval births and adverse outcomes associated with short interval births. The healthcare delivery system also needs to be strengthened to ensure child healthcare services. However, these are challenging for Bangladesh, like other LMICs, for several reasons. The major challenge comes from the facility level, where protective measures, such as providing treatment, are prioritized over preventive measures, such as awareness building. In the current scenario of Bangladesh, mass media and family planning workers are playing an important role in awaring people about the adverse health events, including the importance of longer birth interval, however, the effectiveness of both ways are declining rapidly [66, 67]. The underlying reasons are mismatches between program timing and respondents' availability, unattractive program content and people's interest in other mass media where healthcare messages do not provide adequately [67]. The focus on field-level family planning services and monitoring family planning activities at the field level have also declined over the years [9]. Consequently, respondents' options to get healthcare knowledge and contraception accessibility, which is an important determinant of short birth intervals, have been narrowed over the years [9]. This influences the socio-demographic disadvantaged women mostly, among which births in short interval are highly prevalent [14, 15, 16, 68].

The current study has several strengths and a few limitations. First, the methodological approach used in this study, propensity score matching with inverse probability weighting, helped us to generate the estimate by comparing two equal groups, like a randomized control trial. Second, we analyzed data from two nationally representative households and health facility survey with adequate sample sizes. Third, the covariates were selected using DAGs, where all possible linkages among the covariates are considered. These helped us to generate robust estimates. However, due to the cross-sectional nature of the study, the findings reported are correlational only, rather casual. Another limitation of this study is that though propensity score-based analyses were used, due to non-randomized study design, it may not account for unknown confounders in the same way that a randomized trial does. So the effects of residual confounders may not be avoided. Moreover, though DAG was used to select covariates to be considered in this study, many other covariates, including children food habits, causes of deaths including infection status, society level norms and culture and surrounding environments, were not available in the surveys. However, despite the limitations, this study has novelty, and its findings could help the policy directions in Bangladesh and other LMICs to improve its child, maternal and reproductive health programs.

## Conclusion

SBI was found as the strongest predictors of neonatal, infant and under-five mortality following the adjustment of potential confounders. This indicates that one in every four children in Bangladesh with a short birth interval, are at risk of mortality before reaching their fifth birthday. The SDGs targeted a significant reduction of neonatal and under-five mortality by 2030. With the current high prevalence of SBI in Bangladesh and subsequent mortality risks, the country is at risk of missing the targets unless the prevalence of SBI is addressed by ensuring family planning services and contraception methods are made accessible. Healthcare facility also needs to be strengthened to ensure an adequate preventive and curative cure for mothers and children. Awareness-building programs at the community level about the adverse effects of SBI should also be given priority.

## Acknowledgments

We acknowledge the DHS of the USA for approving us to use their data.

### Ethics approval and consent to participate

The survey analysed was approved by the institutional review board of ICF and the National Research Ethics Committee of the Bangladesh Medical Research Council. Informed consent was obtained from all participants. All necessary patient/participant consent has been obtained and the appropriate institutional forms have been archived. No separate ethical approval was required to conduct this study. We obtained permission to access this survey and conduct this research. All methods were performed in accordance with the relevant guidelines and regulations.

### Consent for publication

Informed consent was obtained from all participants during the survey using a standard DHS consent form.

## Author Contributions

**Conceptualization:** Mohammad Zahidul Islam, Md. Nuruzzaman Khan.

**Formal analysis:** Mohammad Zahidul Islam, Md. Nuruzzaman Khan.

**Supervision:** Md. Mostafizur Rahman, Md. Nuruzzaman Khan.

**Validation:** Md. Mostafizur Rahman.

**Visualization:** Md. Nuruzzaman Khan.

**Writing – original draft:** Mohammad Zahidul Islam, Md. Nuruzzaman Khan.

**Writing – review & editing:** Mohammad Zahidul Islam, Md. Nuruzzaman Khan.

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
