## [Decision Letter · Decision Letter 0]

6 Sep 2022

PONE-D-22-19640Effects of short birth interval on different forms of child mortality in Bangladesh: application of propensity score matching technique with inverse probability of treatment weightingPLOS ONE

Dear Dr. Khan,

Thank you for submitting your manuscript to PLOS ONE. After careful consideration, we feel that it has merit but does not fully meet PLOS ONE’s publication criteria as it currently stands. Therefore, we invite you to submit a revised version of the manuscript that addresses the points raised during the review process.

 Please submit your revised manuscript by October 21, 2022. If you will need more time than this to complete your revisions, please reply to this message or contact the journal office at plosone@plos.org. Please include the following items when submitting your revised manuscript:A rebuttal letter that responds to each point raised by the academic editor and reviewer(s). You should upload this letter as a separate file labeled 'Response to Reviewers'.A marked-up copy of your manuscript that highlights changes made to the original version. You should upload this as a separate file labeled 'Revised Manuscript with Track Changes'.An unmarked version of your revised paper without tracked changes. You should upload this as a separate file labeled 'Manuscript'.If applicable, we recommend that you deposit your laboratory protocols in protocols.io to enhance the reproducibility of your results. Protocols.io assigns your protocol its own identifier (DOI) so that it can be cited independently in the future. For instructions see: https://journals.plos.org/plosone/s/submission-guidelines#loc-laboratory-protocols. Additionally, PLOS ONE offers an option for publishing peer-reviewed Lab Protocol articles, which describe protocols hosted on protocols.io. Read more information on sharing protocols at https://plos.org/protocols?utm_medium=editorial-email&utm_source=authorletters&utm_campaign=protocols.

We look forward to receiving your revised manuscript.

Kind regards,

Betregiorgis Zegeye,

Academic Editor

PLOS ONE

Journal Requirements:

A clean copy of the edited manuscript (uploaded as the new *manuscript* file.

Reviewers' comments:

Reviewer's Responses to Questions

**Comments to the Author**

1. Is the manuscript technically sound, and do the data support the conclusions?

Reviewer #1: Yes

Reviewer #2: Yes

2. Has the statistical analysis been performed appropriately and rigorously? 

Reviewer #1: Yes

Reviewer #2: Yes

3. Have the authors made all data underlying the findings in their manuscript fully available?

Reviewer #1: Yes

Reviewer #2: Yes

4. Is the manuscript presented in an intelligible fashion and written in standard English?

Reviewer #1: Yes

Reviewer #2: Yes

5. Review Comments to the Author

Reviewer #1: The paper worth to publish due to the following criteria:

1-The research is original and its done for first time in Bangladesh

2- The sample size is big enough to show a strong and more accurate result for research at the level of the country

3- The size of control is equal or more than the sample size

4-The statistical requirements can be seen in the required level.

5- The english of manuscript is good and far from mistakes.

Recommendation:

I prefer to write the result is percentage in addition to the writing of the result (number of samples) per 1000. It can be put between brackets.

Reviewer #2: Effects of short birth interval on different forms of child mortality in Bangladesh: application of propensity score matching technique with inverse probability of treatment weighting

Summary and general impression:

The paper made by Mohammad Zahidul Islam et al. team had great and fruitful efforts to find out how SBI affects neonatal, infant, and under-five mortality. The large sample size and the method of analyzing the data using a propensity score matching model; it was wonderful and scientifically sound.

It gave a good insight into the importance of SBI and its impact on the sustainable development goals progress in Bangladesh and the world.

Special issues:

Minor Revision:

1. One of the most important reasons for the death of children under five years of age is the occurrence of infections, accidents and other causes that may not be related in one way or another to death at this age. Has any data been collected on the causes of death for these children, whether from the records or by asking the mothers?

If this is not done, I think it should be placed in limitations section of the study as one of the confounders.

2. The researchers mentioned that they divided the sample according to their residence and mentioned the relative weight of them, but I did not see that in the results, as well as whether there was a difference between the rural and urban residents.

3. Was there an influence of some other important factors such as their social, religious and cultural position on birth spacing?

6. PLOS authors have the option to publish the peer review history of their article (what does this mean?). If published, this will include your full peer review and any attached files.

Reviewer #1: No

Reviewer #2: No

---

## [Author Response · Author response to Decision Letter 0]

5 Oct 2022

Please see the attached response to reviewers file.

---

## [Decision Letter · Decision Letter 1]

12 Dec 2022

PONE-D-22-19640R1Effects of short birth interval on different forms of child mortality in Bangladesh: application of propensity score matching technique with inverse probability of treatment weightingPLOS ONE

Dear Dr. Khan,

Thank you for submitting your manuscript to PLOS ONE. After careful consideration, we feel that it has merit but does not fully meet PLOS ONE’s publication criteria as it currently stands. Therefore, we invite you to submit a revised version of the manuscript that addresses the points raised during the review process.

We look forward to receiving your revised manuscript.

Kind regards,

Betregiorgis Zegeye

Academic Editor

PLOS ONE

Journal Requirements:

Reviewers' comments:

Reviewer's Responses to Questions

**Comments to the Author**

1. If the authors have adequately addressed your comments raised in a previous round of review and you feel that this manuscript is now acceptable for publication, you may indicate that here to bypass the “Comments to the Author” section, enter your conflict of interest statement in the “Confidential to Editor” section, and submit your "Accept" recommendation.

Reviewer #1: All comments have been addressed

Reviewer #3: (No Response)

2. Is the manuscript technically sound, and do the data support the conclusions?

Reviewer #1: Yes

Reviewer #3: Yes

3. Has the statistical analysis been performed appropriately and rigorously? 

Reviewer #1: Yes

Reviewer #3: No

4. Have the authors made all data underlying the findings in their manuscript fully available?

Reviewer #1: Yes

Reviewer #3: Yes

5. Is the manuscript presented in an intelligible fashion and written in standard English?

Reviewer #1: Yes

Reviewer #3: Yes

6. Review Comments to the Author

Reviewer #1: (No Response)

Reviewer #3: Dear Plos ONE team of editorials, thank you for the chance given to me to review a manuscript titled “Retaining Public Health Volunteers beyond COVID-19”. The study has huge importance on retaining volunteers of COVID 19 to other competing natural and artificial disasters including re-emergency of COVID 19 in the Bermingham in Particular and in the globe in general. The following are my comments;

1. The type of the qualitative study was not mentioned. E.g., is that phenomenology, ethnography…etc

2. The study participants were not 360 degrees round included. E.g., where is the view of the government, social service actors, non-government organization…employers and other stakeholders?

3. Who are the respondents? What was your selection criteria’s? Why are they selected? Is that similar or mixed participants?

4. Have you used how many steps of thematic analysis? Where are the codes and categories?

5. You have stated that “…. can be generalized to other scenarios and setting?”. Is this nature of qualitative study?

6. Have you performed stand alone analysis? E.g., email only interviewed participants? Or you have analyzed mixed sources of data? How many FGDs? Criteria’s of FGDs participants? The interview periods? I recommended that you will have two types of results:

i. Face to face interviewed data analysis alone? Its completeness and incompleteness? And what is lacking? So, to email and FGDs?

ii. Mixed source analysis?

7. The findings should be short and concise. Verbatim should be appropriately described. For instance, the age, experience…etc of the participants. You have used many quotations and what are the criteria to use them in your result section?

8. The result and the discussion should be driven from the objective of the study itself.

Regards,

Regards,

7. PLOS authors have the option to publish the peer review history of their article (what does this mean?). If published, this will include your full peer review and any attached files.

Reviewer #1: No

Reviewer #3: No

---

## [Author Response · Author response to Decision Letter 1]

3 Jan 2023

Reviewer #1: (No Response)

Our response: The reviewer did not have any comment to address by us. The comments that reviewer made were for the paper entitled "Retaining Public Health Volunteers beyond COVID-19" that the reviewer clearly mentioned in review. 

Reviewer #3: Dear Plos ONE team of editorials, thank you for the chance given to me to review a manuscript titled “Retaining Public Health Volunteers beyond COVID-19”. The study has huge importance on retaining volunteers of COVID 19 to other competing natural and artificial disasters including re-emergency of COVID 19 in the Bermingham in Particular and in the globe in general. The following are my comments;

1. The type of the qualitative study was not mentioned. E.g., is that phenomenology, ethnography…etc

2. The study participants were not 360 degrees round included. E.g., where is the view of the government, social service actors, non-government organization…employers and other stakeholders?

3. Who are the respondents? What was your selection criteria’s? Why are they selected? Is that similar or mixed participants?

4. Have you used how many steps of thematic analysis? Where are the codes and categories?

5. You have stated that “…. can be generalized to other scenarios and setting?”. Is this nature of qualitative study?

6. Have you performed stand alone analysis? E.g., email only interviewed participants? Or you have analyzed mixed sources of data? How many FGDs? Criteria’s of FGDs participants? The interview periods? I recommended that you will have two types of results:

i. Face to face interviewed data analysis alone? Its completeness and incompleteness? And what is lacking? So, to email and FGDs?

ii. Mixed source analysis?

7. The findings should be short and concise. Verbatim should be appropriately described. For instance, the age, experience…etc of the participants. You have used many quotations and what are the criteria to use them in your result section?

8. The result and the discussion should be driven from the objective of the study itself.

OUR RESPONSE: None of these comments was relevant for our paper. In fact the reviewer made these comments for another paper that the reviewer clearly mentioned in review. Please see the first line of the review that I copy based here too for your look-

 Dear Plos ONE team of editorials, thank you for the chance given to me to review a manuscript titled “Retaining Public Health Volunteers beyond COVID-19”. The study has huge importance on retaining volunteers of COVID 19 to other competing natural and artificial disasters including re-emergency of COVID 19 in the Bermingham in Particular and in the globe in general.

Regards,

---

## [Decision Letter · Decision Letter 2]

20 Mar 2023

PONE-D-22-19640R2Effects of short birth interval on different forms of child mortality in Bangladesh: application of propensity score matching technique with inverse probability of treatment weightingPLOS ONE

Dear Dr. Khan,

Thank you for submitting your manuscript to PLOS ONE. After careful consideration, we feel that it has merit but does not fully meet PLOS ONE’s publication criteria as it currently stands. Therefore, we invite you to submit a revised version of the manuscript that addresses the points raised during the review process.

We look forward to receiving your revised manuscript.

Kind regards,

Betregiorgis Zegeye,

Academic Editor

PLOS ONE

Journal Requirements:

Reviewers' comments:

Reviewer's Responses to Questions

**Comments to the Author**

1. If the authors have adequately addressed your comments raised in a previous round of review and you feel that this manuscript is now acceptable for publication, you may indicate that here to bypass the “Comments to the Author” section, enter your conflict of interest statement in the “Confidential to Editor” section, and submit your "Accept" recommendation.

Reviewer #1: All comments have been addressed

Reviewer #4: (No Response)

Reviewer #5: (No Response)

Reviewer #6: All comments have been addressed

Reviewer #7: All comments have been addressed

Reviewer #8: All comments have been addressed

2. Is the manuscript technically sound, and do the data support the conclusions?

Reviewer #1: Yes

Reviewer #4: Yes

Reviewer #5: Yes

Reviewer #6: Yes

Reviewer #7: Yes

Reviewer #8: Yes

3. Has the statistical analysis been performed appropriately and rigorously? 

Reviewer #1: Yes

Reviewer #4: Yes

Reviewer #5: Yes

Reviewer #6: Yes

Reviewer #7: Yes

Reviewer #8: Yes

4. Have the authors made all data underlying the findings in their manuscript fully available?

Reviewer #1: Yes

Reviewer #4: Yes

Reviewer #5: Yes

Reviewer #6: Yes

Reviewer #7: Yes

Reviewer #8: Yes

5. Is the manuscript presented in an intelligible fashion and written in standard English?

Reviewer #1: Yes

Reviewer #4: Yes

Reviewer #5: Yes

Reviewer #6: Yes

Reviewer #7: Yes

Reviewer #8: Yes

6. Review Comments to the Author

Reviewer #1: The manuscript is sound and worth to publish

The manuscript is corrected and reviewed by me three times in three different times and i think now the manuscript is well prepared and organized

Reviewer #4: The manuscript needs a final check for minor typos, grammar and syntax (marked in track changes below).

1. The work presented meets the requirements for publication. There were some unclear statements such as in the inclusive criteria for women in the survey –ii) passed the most recent night before the survey at the selected households - what does this mean?

2. Please include the WHO interval range to classify SBI in the discussion. Do you think this SBI range is more applicable for Bangladesh context? If yes/no, why?

3. The reference list needs to be checked for missing DOI and the correct format.

Reviewer #5: Thank you for your unreserved deeds and works . It is a crystal clear that the problems under the study is nitty-gritty. Hence, the impotency of the topic both locally and globally unquestionable. As the result, I take this opportunity to say thanks for their unreserved effort and dedication

Reviewer #6: Dear authors,

It was a pleasure reading your article. Thank you for choosing an interesting and essential topic to research.

The article is well written, and I appreciate the previous reviewers' input, which significantly improved your work.

My only suggestion is to add a sentence to the statistical analysis section explaining why you choose the prevalence ratio over the odds ratio in your study.

Keep up the good work,

Mona Abdelrehim

Reviewer #7: Thank you for addressing the points that were highlighted in the previous review. The manuscript has merit.

Reviewer #8: the author has describe the objective of the study clearly, methodology was sound and discussion presented well. good job publishing in this topic. more attention should be done in the region as this also may have significant impact post pandemic

7. PLOS authors have the option to publish the peer review history of their article (what does this mean?). If published, this will include your full peer review and any attached files.

Reviewer #1: No

Reviewer #4: **Yes: **Aminatul Saadiah Abdul Jamil

Reviewer #5: **Yes: **Ibsa Mussa

Reviewer #6: **Yes: **Mona Abdelrehim

Reviewer #7: No

Reviewer #8: No

---

## [Author Response · Author response to Decision Letter 2]

22 Mar 2023

We have added a response to review file as an attachment

---

## [Decision Letter · Decision Letter 3]

10 Apr 2023

Effects of short birth interval on different forms of child mortality in Bangladesh: application of propensity score matching technique with inverse probability of treatment weighting

PONE-D-22-19640R3

Dear Dr. Khan,

We’re pleased to inform you that your manuscript has been judged scientifically suitable for publication and will be formally accepted for publication once it meets all outstanding technical requirements.

Kind regards,

Betregiorgis Zegeye,

Academic Editor

PLOS ONE

Additional Editor Comments (optional):

Reviewers' comments:

Reviewer's Responses to Questions

**Comments to the Author**

1. If the authors have adequately addressed your comments raised in a previous round of review and you feel that this manuscript is now acceptable for publication, you may indicate that here to bypass the “Comments to the Author” section, enter your conflict of interest statement in the “Confidential to Editor” section, and submit your "Accept" recommendation.

Reviewer #4: All comments have been addressed

2. Is the manuscript technically sound, and do the data support the conclusions?

Reviewer #4: Yes

3. Has the statistical analysis been performed appropriately and rigorously? 

Reviewer #4: Yes

4. Have the authors made all data underlying the findings in their manuscript fully available?

Reviewer #4: Yes

5. Is the manuscript presented in an intelligible fashion and written in standard English?

Reviewer #4: Yes

6. Review Comments to the Author

Reviewer #4: Thank you very much for your hard work and dedication in improving the paper after several rounds of reviews. The result is a polished manuscript to the standard expected. The study has novelty and could add to our understanding of the repercussions of SBI on child mortality.

7. PLOS authors have the option to publish the peer review history of their article (what does this mean?). If published, this will include your full peer review and any attached files.

Reviewer #4: No

---

## [Editor Report · Acceptance letter]

12 Apr 2023

PONE-D-22-19640R3 

Effects of short birth interval on different forms of child mortality in Bangladesh: application of propensity score matching technique with inverse probability of treatment weighting 

Dear Dr. Khan:

I'm pleased to inform you that your manuscript has been deemed suitable for publication in PLOS ONE. Congratulations! Your manuscript is now with our production department. 

Kind regards, 

on behalf of

Mr. Betregiorgis Hailu Zegeye 

Academic Editor

PLOS ONE